# Is Osteopontin a Reliable Biomarker for Endometriosis?

**DOI:** 10.3390/ijms252211857

**Published:** 2024-11-05

**Authors:** Aleksandra Zygula, Kamil Kiecka, Anna Sankiewicz, Mariusz Kuzmicki, Michal Ciebiera, Tadeusz Issat, Wojciech Drygas, Krzysztof Cendrowski, Ewa Gorodkiewicz, Piotr Laudanski

**Affiliations:** 1OVIklinika Infertility Center, 01-377 Warsaw, Poland; 2Faculty of Medical Sciences in Katowice, Medical University of Silesia, 40-055 Katowice, Poland; kamil.kiec2001@gmail.com; 3Angelius Provita Hospital, 40-611 Katowice, Poland; 4Bioanalysis Laboratory, Faculty of Chemistry, University of Bialystok, 15-328 Bialystok, Poland; ania@uwb.edu.pl (A.S.); ewka@uwb.edu.pl (E.G.); 5Department of Gynecology and Gynecological Oncology, Medical University of Bialystok, 15-089 Bialystok, Poland; mariusz.kuzmicki@umb.edu.pl; 6Second Department of Obstetrics and Gynecology, Center of Postgraduate Medical Education, 00-189 Warsaw, Poland; michal.ciebiera@gmail.com; 7Warsaw Institute of Women’s Health, 00-189 Warsaw, Poland; 8Development and Research Center of Non-Invasive Therapies, Pro-Familia Hospital, 35-302 Rzeszow, Poland; 9Department of Obstetrics and Gynecology, Institute of Mother and Child in Warsaw, 01-211 Warsaw, Poland; tadeusz.issat@imid.med.pl; 10World Institute for Family Health, Calisia University, 62-800 Kalisz, Poland; w.drygas@uniwersytetkaliski.edu.pl; 11Department of Obstetrics, Gynecology and Gynecological Oncology, Medical University of Warsaw, 02-091 Warsaw, Poland; krzysztof.cendrowski@wum.edu.pl; 12Women’s Health Research Institute, Calisia University, 62-800 Kalisz, Poland

**Keywords:** endometriosis, osteopontin, plasma, peritoneal fluid, infertility, SPRI, diagnosis

## Abstract

This study aimed to evaluate the concentration of osteopontin in peritoneal fluid and plasma as potential biomarkers for diagnosing endometriosis. Osteopontin levels were measured using surface plasmon resonance imaging (SPRI) biosensors in patients suspected of having endometriosis. Plasma samples were collected from 120 patients, and peritoneal fluid was collected from 86 patients. Based on the detection of endometriosis lesions during laparoscopy, participants were divided into a study group (patients with endometriosis) and a control group (patients without endometriosis). The results showed no significant differences in plasma osteopontin levels between women with endometriosis and the control group (19.86 ± 6.72 ng/mL vs. 18.39 ± 4.46 ng/mL, *p* = 0.15). Similarly, peritoneal fluid osteopontin concentrations did not differ significantly between patients with and without endometriosis (19.04 ± 5.37 ng/mL vs. 17.87 ± 5.13 ng/mL, *p* = 0.29). Furthermore, osteopontin levels in both plasma and peritoneal fluid were not significantly associated with the stage of endometriosis, the presence of endometrioma, or the menstrual cycle phase. The findings of this study do not support osteopontin concentration as a reliable biomarker for endometriosis. However, further research is necessary to explore osteopontin’s potential role in the disease.

## 1. Introduction

Endometriosis is a common chronic gynecological condition that has a substantial impact on various aspects of women’s lives [1]. It is defined by the presence of endometrial-like tissue growing outside the uterine cavity. The condition affects 4–15% of the general female population, with prevalence rates rising to as high as 50% among women experiencing infertility [2,3,4]. Despite its widespread prevalence, the true rate of endometriosis is likely underestimated due to its often subclinical nature and the nonspecific presentation of its symptoms.

Women with endometriosis often experience severe menstrual pain (dysmenorrhea), pain during intercourse (dyspareunia), and chronic pelvic pain (CPP), with these symptoms typically emerging during reproductive years [5,6]. The condition can present in several forms—peritoneal endometriosis, ovarian endometrioma, and deep infiltrating endometriosis—each contributing to a wide array of clinical symptoms, including infertility anxiety, and depression [1]. Both earlier and recent studies indicate that endometriosis approximately doubles the risk of infertility, further underscoring its impact on reproductive health [7,8].

Diagnosing endometriosis is especially challenging due to its variable and nonspecific symptoms, resulting in an average diagnostic delay of 7–12 years [4,9]. This prolonged delay highlights the critical need for continued research to identify reliable, non-invasive biomarkers for early detection, which could greatly enhance patient outcomes and overall quality of life [10].

Despite extensive research, the exact pathogenesis of endometriosis remains unclear. Sampson’s theory, which is widely recognized, suggests that retrograde menstruation plays a significant role [11]. However, given that around 90% of menstruating women experience retrograde menstruation but only about 10% develop endometriosis, additional factors are likely involved [12]. Other theories propose genetic, environmental, and immunological contributions to disease development [4]. Disrupted immune regulation, along with inflammation involving various immune cells, cytokines, chemokines, metalloproteinases, and cathepsins, has been implicated in the development, invasion, and angiogenesis of ectopic lesions [13,14,15,16,17,18,19]. Cytokines such as IL-37, IL-17A, IL-10, and IL-2 play essential roles in modulating immune responses in endometriosis [20]. Higher levels of serum IL-37 and IL-10, and significantly lower levels of serum IL-17A and IL-2 were detected in patients with endometriosis [20]. Additionally, studies indicate that IL-6 levels correlate with the presence and severity of pelvic endometriosis, underscoring its role as a key mediator of inflammation and a potential marker of disease severity [21].

Beyond cytokines, metabolic markers such as ghrelin, glucagon-like peptide 1 (GLP-1), glucagon, and visfatin have been examined in relation to endometriosis. Reduction in these markers in the peritoneal fluid of endometriosis patients suggests they may contribute to disease progression by promoting pro-inflammatory activity, particularly through their effects on peritoneal macrophages [22].

Leptin, a hormone linked to inflammation and energy metabolism, has also been extensively studied in the context of endometriosis. Although findings are somewhat ambiguous, most research suggests that leptin concentrations are elevated in both peritoneal and follicular fluid of women with endometriosis compared to controls [23,24].

However, these findings alone do not fully explain the complex pathogenesis of endometriosis. The molecular mechanisms underlying the metabolic processes involved in the disease remain poorly understood [25]. Recently, molecules like osteopontin (OPN), which play roles in regulating various physiological and pathological processes, have been evaluated as potential biomarkers for endometriosis [26]. OPN, encoded by the *SPP1* gene and also known as early T-lymphocyte activation 1 protein (ETA1), belongs to the small integrin-binding ligand N-linked glycolprotein (SIBLING) family [27]. The term osteopontin derives from “osteo”, meaning bone, and “pontin”, meaning bridge, reflecting its role in connecting bone cells to the bone extracellular matrix [27]. This glycoprotein is produced by a variety of cells, including osteoclasts, osteoblasts, epithelial cells, endothelial cells, neurons, and immune cells such as T cells, NK cells, macrophages, and Kupffer cells [28]. OPN is consistently present in multiple tissues, including the kidney, breast, brain, skin, bone, bone marrow, and bladder, as well as in bodily fluids like plasma, urine, milk, and bile [28,29].

OPN is a multifunctional glycoprotein involved in various physiological and pathological processes, such as mineralization, angiogenesis, immune regulation, and inflammation [30]. It recruits monocytes and macrophages, mediates cytokine secretion in leukocytes [31], and plays a critical role in blastocyst-endometrium adhesion in the endometrium during the window of implantation (WOI) via interaction with αvβ integrin [32].

OPN is implicated in cancer progression, neoplastic transformation, and tissue remodeling across several cancers, notably ovarian, colorectal, and hepatocellular carcinoma (HCC) [33,34,35]. In ovarian cancer, it promotes tumor progression by enhancing cell migration, invasion, and survival, with elevated levels linked to poor prognosis and metastasis. In colorectal cancer, OPN supports tumor growth through angiogenesis and immune evasion, correlating with advanced disease and reduced survival rates. In HCC, it facilitates tumor growth and resistance to apoptosis, where high expression indicates poorer outcomes. OPN also regulates angiogenesis, cell adhesion, apoptosis, and inflammation, which impact cell attachment, migration, invasion, and proliferation across these cancers [31]. Beyond oncology, OPN plays essential roles in immune response regulation, particularly by recruiting immune cells and promoting cytokine secretion in chronic inflammatory conditions. It has emerged as a biomarker for early Alzheimer’s disease due to its influence on neuroinflammation and neuronal survival through microglial activity.

Additionally, OPN is crucial for bone health, facilitating communication between osteoclasts and osteoblasts to regulate bone resorption and formation. Furthermore, OPN contributes to vascular calcification by promoting inflammation and plaque formation in atherosclerosis, advancing heart disease [36]. Expressed by immune cells, it impacts calcification and may serve as a therapeutic target. These diverse functions underscore OPN’s potential as both a therapeutic target and biomarker in a range of conditions, including cancer, chronic inflammation, neurodegeneration, and reproductive diseases. Moreover, OPN’s roles in cell migration, attachment, and invasion suggest a potential link to the pathogenesis of endometriosis through mechanisms similar to those observed in cancer, indicating that OPN could serve as a valuable biomarker for diagnosing or monitoring endometriosis

The aim of this study was to compare OPN levels in biological fluids—plasma and peritoneal fluid—between women with endometriosis and a control group, using surface plasmon resonance imaging (SPRi) biosensors. Additionally, this study aimed to evaluate the potential of OPN as a biomarker for endometriosis

SPRi stands at the cutting edge of optical sensing technologies, offering real-time, direct, and label-free detection and monitoring of biomolecular interactions [37]. This technique uses the surface of a metal to generate a resonance effect of the metal’s surface electrons when exposed to polarized light radiation. This phenomenon is sensitive to changes at the metal surface, enabling the analysis of cells and surface-associated biomolecules. A significant advantage of SPRi is its molecular specificity and its highly sensitive measurement of the refractive index, making it ideal for quantifying biomolecules, such as various proteins [37,38,39,40,41,42,43]. The application of stationary SPRi in model investigations and the determination of biomarkers in clinical samples has shown that the technique is effective without needing signal enhancement or analyte preconcentration. Recent reviews have recognized SPRi detection as the leading method among surface plasmon-based techniques.

## 2. Results

Table 1 and Table 2 present the characteristics of the patients from whom plasma and peritoneal fluid were collected, respectively. Both groups were homogeneous and adequately matched. A significant statistical difference was found between the groups when analyzing peritoneal fluid: the incidence of primary infertility was significantly higher in the study group compared to the control group (46.94% vs. 27.50%, *p*-value 0.0302). In the group where plasma was analyzed, a statistically significant difference was observed in the average BMI, which was lower in the study group compared to the control group (21.565 kg/m^2^ vs. 22.739 kg/m^2^, respectively).

The levels of osteopontin in peritoneal fluid (ng/mL) did not differ significantly between the groups, measuring 19.04 ± 5.37ng/mL in the study group and 17.87 ± 5.130 ng/mL in the control group (Table 3).

The mean plasma osteopontin concentration (ng/mL) was comparable in the two groups, with 19.86 ± 6.72 ng/mL in women with endometriosis and 18.39 ± 4.46 ng/mL in the control group (Table 3).

There were no significant differences observed in the osteopontin level both in plasma and peritoneal fluid, based on the endometriosis stage, presence of endometrioma, or menstrual cycle phase (Table 4).

The effect sizes for the comparisons under study were low, ranging from 0.02 to less than 0.01.

Based on the presented data, osteopontin levels in both peritoneal fluid and plasma exhibited limited predictive value for endometriosis (Figure 1 and Figure 2). In both cases, the lower bounds of the 95% CI for AUC remained below 0.5, and the ROC curve indicated that both variables acted as stimulants in some intervals and as destimulants in others. Consequently, the result was not statistically significant (Figure 1 and Figure 2).

## 3. Discussion

OPN, a multifunctional cytokine, is involved in various biological processes, including angiogenesis, cell adhesion, apoptosis, and inflammation—all mechanisms that underlie endometriosis pathogenesis [31,44]. This suggests that it may play a role in the disease’s development. However, research investigating OPN as a potential biomarker for endometriosis has yielded conflicting results. This study is the first to demonstrate that plasma and peritoneal OPN concentration levels do not differ significantly between women with endometriosis and healthy controls. This result contrasts with prior research; for instance, studies by D’Amico et al. and Cao et al. reported elevated plasma OPN levels in women with endometriosis [45,46].

Conversely, other studies have reported reduced OPN levels, underscoring the inconsistency in findings related to OPN concentrations in endometriosis. Supporting this, Moszynski et al. found that median serum OPN levels were significantly lower in women with endometriosis compared to those with other benign ovarian conditions. [47]. Notably, none of these studies have examined OPN levels in peritoneal fluid, as we have in the present study. Peritoneal fluid in women with endometriosis is unique because it is in direct contact with endometriotic lesions, making it a promising medium for detecting potential biomarkers. Measuring OPN levels in peritoneal fluid could provide a more accurate assessment of its potential as a biomarker, reflecting local pathological processes more closely than systemic fluids.

Moreover, studies on OPN expression in the eutopic endometrium have yielded mixed results. Yang et al. identified elevated OPN expression in the eutopic endometrium of women with endometriosis, a finding further supported by D’Amico et al., who also noted significantly higher mRNA expression of the OPN gene [45,48,49]. Furthermore, Hapangama et al. reported elevated OPN expression in the eutopic endometrium, specifically during the luteal phase of the menstrual cycle in women with endometriosis [50]. Conversely, Cho et al. also observed elevated OPN mRNA expression in women with endometriosis during both the proliferative and secretory phases [49]. Additionally, Casals et al. found no difference in OPN concentration in the endometrium between women with minimal to mild endometriosis and controls. In contrast, Wei et al. observed decreased expression of OPN during the late luteal phase in women with r-AFS stages I to IV endometriosis [51,52]. In the present study, no differences in OPN levels were observed between the proliferative and secretory phases in either plasma or peritoneal fluid. Although this study did not assess OPN levels in the eutopic endometrium, future research will aim to explore this area.

Interestingly, our analysis of OPN levels by endometriosis type yielded unexpected results. Previous studies have shown significantly lower OPN levels in women with deeply infiltrative endometriosis (DIE) compared to those with superficial peritoneal endometriosis (SUP), as well as reduced levels in r-AFS stages II and IV compared to stage I [53]. This suggests that OPN may play a role in producing pro-inflammatory cytokines by epithelial cells and/or macrophages, particularly during the early immune response in peritoneal endometriosis [53]. Additionally, OPN may influence the spread of peritoneal lesions, pointing to a more prominent involvement in the peritoneal form of endometriosis than in DIE [53]. In the present study, however, we found no significant difference in OPN levels between r-AFS stages I-II and stages III-IV. Future research is needed to clarify whether OPN levels vary with disease stage or if OPN concentrations are consistently higher in women with superficial peritoneal endometriosis compared to healthy controls.

A recent breakthrough study by Albu et al. examined OPN levels in the serum of women with endometriosis in response to progestagen treatment and surgical intervention [54]. Their results suggest that while OPN responds to these interventions, it serves as a relatively weak biomarker for endometriosis. The findings demonstrated that in patients treated with progestagens, either before or after surgery, serum OPN levels increased. Patients who underwent surgery alone had low OPN levels six months post-operation, whereas those who received oral desogestrel after surgery exhibited a mean OPN level 10.31 times higher than those without progestagen treatment and significantly higher than the control group. These findings are intriguing, as they suggest that OPN levels may be more influenced by progestagen treatment than by the presence of endometriosis itself. This observation raises important questions about the role of progestagens in modulating OPN levels, potentially overshadowing the direct impact of endometriosis on this biomarker. Thus, hormonal treatment may induce a substantial increase in OPN, complicating its reliability as a diagnostic or monitoring tool for endometriosis. Further research is needed to disentangle the effects of progestagens from those of endometriosis to better understand the underlying mechanisms and improve the clinical utility of OPN measurements.

Our results support the hypothesis that the significant increase in OPN may be induced by hormonal treatment rather than endometriosis itself. In the present study, we excluded patients who had undergone hormonal treatment within three months before laparoscopy, which could explain why we did not observe a significant difference in OPN levels. This exclusion criterion was implemented to eliminate any potential confounding effects of recent hormonal therapy on OPN measurements, ensuring that the observed OPN levels were more reflective of the underlying endometriosis rather than the influence of exogenous hormones. These findings underscore the importance of considering the timing and presence of hormonal treatments when evaluating OPN as a biomarker for endometriosis.

The absence of significant differences in OPN levels in plasma and peritoneal fluid suggests that, contrary to initial hypotheses, OPN may not be a reliable systemic marker for diagnosing or monitoring the progression of endometriosis. This finding indicates that OPN’s role in endometriosis may be more localized, potentially limited to the endometrial tissue itself. While OPN could be elevated within endometrial tissue due to inflammation or cellular responses, these increases may not trigger systemic changes detectable in peripheral biological fluids. This compartmentalization suggests that OPN plays a crucial role within the microenvironment of endometriotic tissue—impacting local inflammation, cell adhesion, and tissue remodeling—without a significant release into the bloodstream or peritoneal cavity. Consequently, this highlights the need to reconsider OPN as a systemic biomarker for endometriosis and suggests that other markers, or a combination of markers, might offer more reliable options for systemic detection and assessment.

We recognize certain limitations in our study. The relatively small sample size of both the study and control groups is a notable constraint. To address this, we have planned further studies involving a multicenter patient population in Poland to broaden and deepen the analysis.

One of the strengths of our study is the use of SPRI, a well-established and reliable measurement technology. Compared to traditional methods of OPN detection, SPRI biosensors do not require sample processing and have lower restrictions on the volume of biological material and operating conditions. Additionally, by assessing OPN concentrations in both plasma and peritoneal fluid, we explored their interrelationships and potential impact on the pathogenesis of endometriosis. Another advantage of our research is the meticulous sampling procedure, which ensured the purity of the collected peritoneal fluid. Importantly, our study adds to a series of publications investigating the potential role of specific molecules as biomarkers for endometriosis [24,55,56,57].

## 4. Materials and Methods

### 4.1. Study Population

This multicenter, cross-sectional study was carried out across eight Departments of Obstetrics and Gynecology in Poland between 2018 and 2019. Detailed recruitment information is available in our recent publication [43]. All participants, including both patients with endometriosis and controls, provided written informed consent, and the study was approved by the Ethics Committee of the Medical University of Warsaw (KB/223/2017).

The study group comprised individuals aged 19 to 45 years who were scheduled for laparoscopic surgeries due to one or more non-malignant conditions, such as infertility, chronic pelvic pain syndrome, ovarian cysts, or suspected endometriosis. Exclusion criteria included irregular menstrual cycles (less than 25 days or more than 35 days), recent hormonal treatment within three months before laparoscopy, previous or current pelvic inflammatory disease, uterine fibroids, polycystic ovary syndrome, autoimmune comorbidities, malignancies, or any prior history of surgical treatment. All patients underwent a gynecological examination and vaginal ultrasonography before surgery. Routine additional radiological examinations to evaluate extraperitoneal endometriosis were not included in the standard procedure.

Each patient was evaluated based on the revised American Fertility Society (AFS) classification of endometriosis, supported by histological examination of collected specimens. Additionally, all patients completed a World Endometriosis Research Foundation (WERF) clinical questionnaire. Controls were patients without visible endometriosis during laparoscopy.

The menstrual cycle phase was determined based on the last menstrual period and the average length of the menstrual cycle. To establish the phases of the menstrual cycle in both women with and without endometriosis, histological dating of eutopic endometrial samples was conducted simultaneously with the collection of pathological lesions. The details of the collection and structure of the study groups have been previously published [43]. Appendix A contains detailed information on the collection and structure of the study groups.

Patients diagnosed with endometriosis during laparoscopy were categorized into corresponding endometriosis stage subgroups (I–IV). OPN concentration assessment involved the collection of plasma and peritoneal fluid, following the Endometriosis Phenome and Biobanking Harmonisation Project’s standard operating procedures [58]. After excluding outlier results, the final analysis included 120 plasma samples (67 from patients with endometriosis and 53 from controls) and 86 peritoneal fluid samples (48 from patients with endometriosis and 38 from controls).

### 4.2. Method of Measuring Osteopontin Concentration

In this study, osteopontin concentrations in plasma and peritoneal fluid were measured using a Surface Plasmon Resonance apparatus in the imaging version, as described in detail in Appendix B.

### 4.3. Statistical Analyses

Statistical analysis was performed using the statistical package Statistica 13 (TIBCO Software Inc., Palo Alto, CA, USA; 2017).

Measures of central tendency and variability were calculated in the control and study subgroups. Compliance of the distributions of OPN concentrations with the normal distribution was tested using the Shapiro–Wilk test. The distribution of OPN concentrations in both peritoneal fluid and plasma was consistent with normal; after checking the assumption of homogeneity of variances in the subgroups, an appropriate *t*-test was used (in the absence of homogeneity of variances, it was a *t*-test with the Cochran–Cox correction) and the effect size was based on η^2^ (eta squared).

A significance level α = 0.05 was adopted in this study and was used to estimate the *p*-value.

## 5. Conclusions

In conclusion, OPN’s role as a reliable endometriosis biomarker remains controversial. Our study aligns with the thesis that peritoneal fluid and serum OPN levels may not significantly differ between patients with endometriosis and healthy individuals.

Hence, additional research is imperative to elucidate the role of OPN in endometriosis. Future studies should explore the impact of various hormonal regimens and their withdrawal periods on OPN levels to refine their application in clinical practice. Further research is needed to delve into the underlying mechanisms through which OPN operates in endometriosis, determining whether the changes in OPN concentrations observed by some investigators result from the disease’s pathogenesis. Additionally, more comprehensive data regarding OPN as a diagnostic biomarker are necessary for a clearer understanding of its potential utility in the diagnostic process of endometriosis.

## Figures and Tables

**Figure 1 ijms-25-11857-f001:**
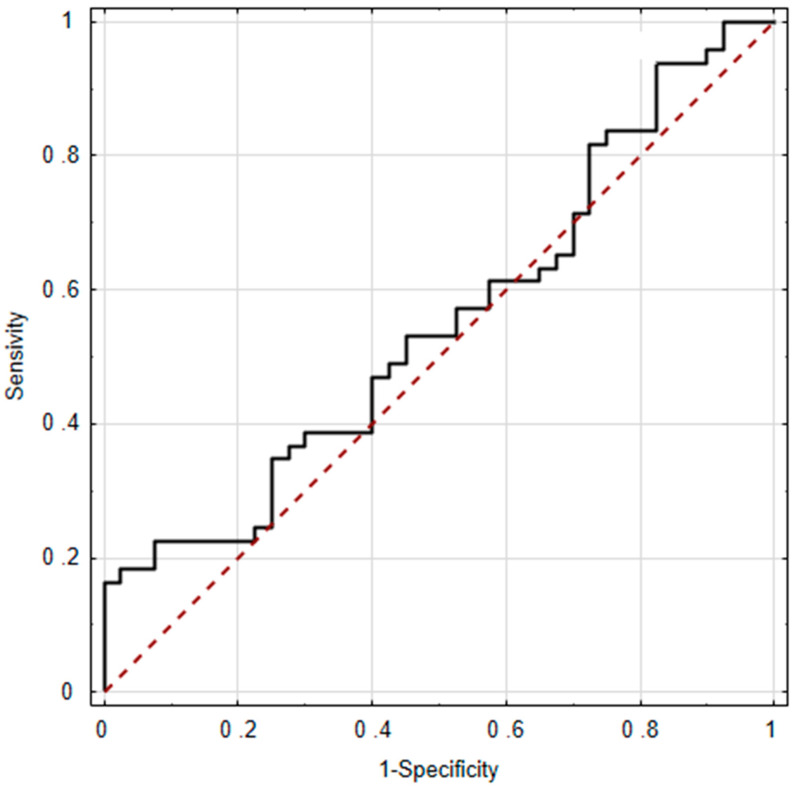
Osteopontin in peritoneal fluid as a predictor of endometriosis AUC = 0.551; ±95% CI (0.430–0.671); *p* = 0.4108.

**Figure 2 ijms-25-11857-f002:**
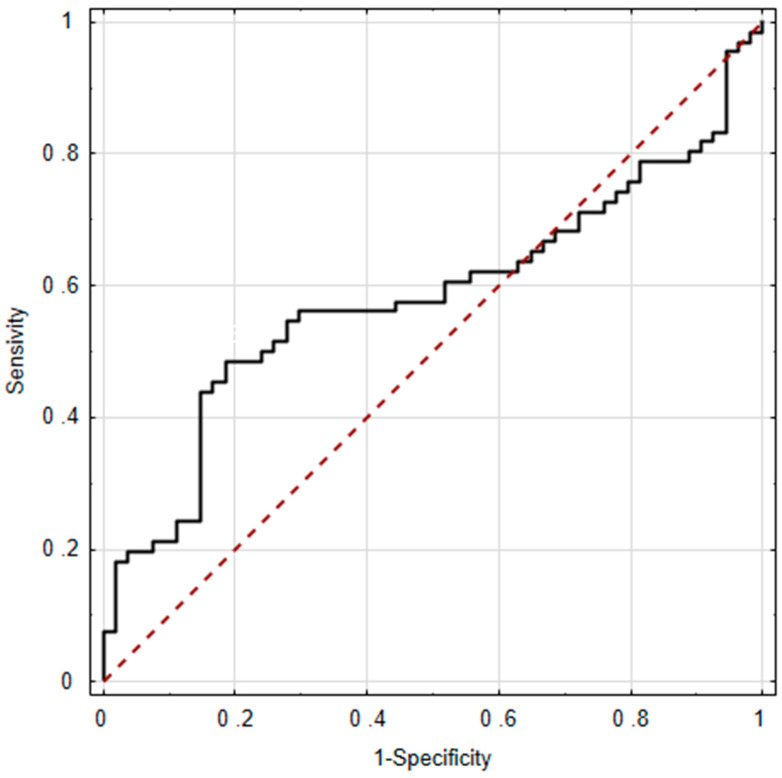
Osteopontin in plasma as a predictor of endometriosis AUC = 0.582; ±95% CI (0.479–0.686); *p* = 0.1196.

**Table 1 ijms-25-11857-t001:** Baseline characteristics of participants in whom peritoneal fluid was collected.

Variable	N	Endometriosis (M ± SD)	N	Controls (M ± SD)	*p*-Value
Age [years]	48	31.90 ± 4.50	38	30.42 ± 6.52; 38	0.23
BMI [kg/m^2^] (Me ± IQR)	47	21.967 ± 2.974	34	21.754 ± 4.828; 34	0.55
N/A	1	--	4	--	--
	**N**	**%**	**N**	**%**	
rAFS stage of disease					
I	14	31.11	--	--	--
II	7	15.56	--	--	--
III	18	40.00	--	--	--
IV	6	13.33	--	--	--
Presence of endometrioma (%)	27	55.1	--	--	--
Presence of primary infertility (%)	23	46.94	11	27.50	0.03
Presence of secondary infertility (%)	5	10.20	6	15.00	0.24
Menstrual cycle phase:					
Proliferative	30	62.50	29	72.50	0.32
Luteal	18	37.50	11	27.50	

Abbreviations: N, sample size; BMI, body mass index; N/A, not applicable. Numerical data are presented as Me (median), IQR (interquartile range), or M (mean) ± SD (standard deviation) depending on the distribution.

**Table 2 ijms-25-11857-t002:** Baseline characteristics of participants in whom plasma was collected.

Variable	N	Endometriosis	N	Controls	*p*-Value
Age [year] (M ± SD; N)	67	32.99 ± 4.70	53	32.057 ± 6.50	0.82
BMI [kg/m^2^] (Me ± IQR; N)	61	21.565 ± 3.09	50	22.739 ± 5.30	0.04
N/A	6	--	3	--	--
	**N**	**%**			
rAFS stage of disease %. N					
I	22	34.38	--	--	--
II	11	17.19	--	--	--
III	22	34.38	--	--	--
IV	9	14.06	--	--	
Presence of endometrioma	36	53.73		--	--
Presence of primary infertility (%)	31	46.27	20	37.03	0.15
Presence of secondary infertility (%)	7	10.45	8	14.81	0.23
Menstrual cycle phase (Me ± IQR; N)					
Proliferative	47	47 ± 71.21	41	41 ± 75.93	0.56
Luteal	19	19 ± 28.79	13	13 ± 24.08	

Abbreviations: N, sample size; BMI, body mass index; N/A, not applicable. Numerical data are presented as Me (median), IQR (interquartile range), or M (mean) ± SD (standard deviation) depending on the distribution.

**Table 3 ijms-25-11857-t003:** Osteopontin level in body fluids [ng/mL].

Factors	N	M	−95% CI	+95% CI	Min.	Max.	SD	*t*	*p*	η^2^
Peritoneal fluid										
Endometriosis	49	19.04	17.50	20.58	8.36	32.75	5.37	−1.04	0.29	0.01
Control	40	17.87	16.23	19.51	6.99	25.20	5.130			
Plasma										
Endometriosis	66	19.86	18.21	21.51	6.02	37.13	6.72	−1.43	0.15	0.02
Control	54	18.39	17.17	19.61	6.34	29.51	4.46			

Abbreviations: N, sample size; BMI, body mass index; N/A, not applicable. Numerical data are presented as M (mean) ± SD (standard deviation) and CI (confidence interval). η^2^ Eta squared.

**Table 4 ijms-25-11857-t004:** Osteopotin concentration [ng/mL] according to the stage of endometriosis, endometrioma, and cycle phase.

Factors		N	M	−95% CI	+95% CI	Min.	Maks.	SD	*t*	*p*	η^2^
rAFS stage of disease											
Peritoneal fluid	I–II	21	18.27	15.62	20.91	9.10	32.75	5.81	−1.10	0.27	0.03
	III–IV	24	20.05	17.90	22.21	8.36	27.03	5.10			
Plasma	I–II	33	19.54	17.41	21.67	6.02	32.39	6.000	−0.56	0.57	<0.01
	III–IV	30	20.51	17.67	23.34	7.68	37.13	7.59			
Endometrioma											
Peritoneal fluid	Yes	27	19.40	17.30	21.50	8.36	27.03	5.31	−1.05	0.29	0.01
	No	62	18.13	16.80	19.46	6.99	32.75	5.24			
Plasma	Yes	35	20.12	17.59	22.65	7.68	37.13	7.36	−1.11	0.26	0.01
	No	85	18.82	17.72	19.91	6.02	32.39	5.08			
Cycle phase											
Peritoneal fluid	Proliferative	59	18.34	16.87	19.80	6.99	32.75	5.62	−0.56	0.57	<0.01
	Secretive	29	19.01	17.28	20.74	8.36	25.89	4.55			
Plasma	Proliferative	87	19.71	18.42	20.99	6.019	37.13	6.03	1.35	0.17	0.02
	Secretive	32	18.09	16.27	19.91	9.04	32.39	5.04			

Abbreviations: N, sample size; BMI, body mass index; N/A, not applicable. Numerical data are presented as M (mean) ± SD (standard deviation) and CI (confidence interval). η^2^ Eta squared.

## Data Availability

The raw data supporting the conclusions of this article will be made available by the authors on request.

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
