# Peer review of "Is Osteopontin a Reliable Biomarker for Endometriosis?"

_ijms, 2024, doi:10.3390/ijms252211857_

Round 1
Reviewer 1 Report
Comments and Suggestions for Authors
Considering that endometrial osteopontin mRNA expression and plasma osteopontin levels are increased in patients with endometriosis, please analyze further your “negative” findings.
Comments on the Quality of English LanguagePlease, use the abbreviation OPN, the FIRST time osteopontin appears in the text.
Author Response
Comments 1: Considering that endometrial osteopontin mRNA expression and plasma osteopontin levels are increased in patients with endometriosis, please analyze further your “negative” findings.
Thank you for pointing this out. We agree with this comment. Therefore we add analysis of "negative" findings in the disscussion, lines 270 - 282:
The absence of significant differences in osteopontin levels in plasma and peritoneal fluid suggests that, contrary to initial hypotheses, osteopontin may not be a reliable systemic marker for diagnosing or monitoring the progression of endometriosis. This finding indicates that osteopontin’s role in endometriosis may be more localized, potentially limited to the endometrial tissue itself. While osteopontin could be elevated within endometrial tissue due to inflammation or cellular responses, these increases may not trigger systemic changes detectable in peripheral biological fluids. This compartmentalization suggests that osteopontin plays a crucial role within the microenvironment of endometriotic tissue—impacting local inflammation, cell adhesion, and tissue remodeling—without significant release into the bloodstream or peritoneal cavity. Consequently, this highlights the need to reconsider osteopontin as a systemic biomarker for endometriosis and suggests that other markers, or a combination of markers, might offer more reliable options for systemic detection and assessment.
Comment 2: The English could be improved to more clearly express the research.
We have carefully revised the English throughout the document to enhance clarity and ensure our research findings are presented more effectively. We hope this updated language meets Your expectations.
Comment 3:
Please, use the abbreviation OPN, the FIRST time osteopontin appears in the text.
We introduce the abbreviation "OPN" the first time "osteopontin" appears in the text.
Reviewer 2 Report
Comments and Suggestions for Authors
There has been a boom in osteopontin research in recent years, and the submitted work adds to the body of evidence on the roles of osteopontin in gynecology. This study aligns with a systematic exploration of peritoneal fluid markers of endometriosis previously published by some of the authors. The main goal of the submitted work was to analyze the concentrations of osteopontin in peritoneal fluid and plasma and to evaluate their utility as endometriosis biomarkers. I agree with this approach, as it provides relevant pathophysiological insights into both endometriosis and osteopontin.
Research on various molecules (e.g., cytokines; see PMID: 29308924, PMID: 28962161) in the peritoneal fluid of endometriosis patients is not new. The authors primarily reference their own commendable research to support this point. However, broadening the perspective by including additional references (PMID: 36142272, PMID: 34126469) would be advisable.
Additionally, the discussion should cover recent advances in osteopontin research beyond endometriosis, such as in ovarian cancer (PMID: 35406551), colorectal cancer (PMID: 39410512), and hepatocellular carcinoma (PMID: 39398562), as well as its physiological roles outside the pelvis (PMID: 39344228, PMID: 39072932, PMID: 38919629).
Tables 1 and 2: The accumulation of numbers makes the tables difficult to read (e.g., "Age [yr] (M ± SD; N) 31.90 ± 4.50; 48"). I suggest listing the numbers (n) separately for better clarity.
Additionally, in Table 1, the meaning of the variable "Presence of endometrioma 27 ± 55.10" is unclear without further explanation. Does it refer to the diameter of the endometrioma, the age of the patients, or the percentage of patients with endometrioma?
The numbers for controls in the rows "age" (n = 38) and "menstrual cycle" (n = 40) are inconsistent. If data are missing, please indicate this as "N/A."
Table 3: Abbreviations are used without explanation (e.g., -95% PU, +95% PU). Please provide definitions for all abbreviations.
Was the rAFS staging system the only classification used? The #Enzian classification is currently the most widely used and comprehensive system for endometriosis in Europe. If this information is available, please include the #Enzian scoring for patient characteristics.
Author Response
Comment 1:Research on various molecules (e.g., cytokines; see PMID: 29308924, PMID: 28962161) in the peritoneal fluid of endometriosis patients is not new. The authors primarily reference their own commendable research to support this point. However, broadening the perspective by including additional references (PMID: 36142272, PMID: 34126469) would be advisable.
Response 1:
Thank you for the comment. We provide additional information on various molecules involved in the pathogenesis of endometriosis, in the introduction part line 77-91:
Cytokines such as IL-37, IL-17A, IL-10, and IL-2 play essential roles in modulating immune responses in endometriosis . Higher levels of serum IL-37 and IL-10, and significantly lower levels of serum IL-17A and IL-2 were detected in patients with endometriosis. Additionally, studies indicate that IL-6 levels correlate with the presence and severity of pelvic endometriosis, underscoring its role as a key mediator of inflammation and a potential marker of disease severity.
Beyond cytokines, metabolic markers such as ghrelin, GLP-1, glucagon, and visfatin have been examined in relation to endometriosis. The reduction of these markers in the peritoneal fluid of endometriosis patients suggests they may contribute to disease progression by promoting pro-inflammatory activity, particularly through their effects on peritoneal macrophages.
Leptin, a hormone linked to inflammation and energy metabolism, has also been extensively studied in the context of endometriosis. Although findings are somewhat ambiguous, most research suggests that leptin concentrations are elevated in both peritoneal and follicular fluid of women with endometriosis compared to controls.
Comment 2:Additionally, the discussion should cover recent advances in osteopontin research beyond endometriosis, such as in ovarian cancer (PMID: 35406551), colorectal cancer (PMID: 39410512), and hepatocellular carcinoma (PMID: 39398562), as well as its physiological roles outside the pelvis (PMID: 39344228, PMID: 39072932, PMID: 38919629).
Response 2:
We have added additional information on the role of OPN in both physiological and pathological processes beyond endometriosis - line 106 - 136.
Osteopontin (OPN) is a multifunctional glycoprotein involved in various physiological and pathological processes, such as mineralization, angiogenesis, immune regulation, and inflammation. It recruits monocytes and macrophages, mediates cytokine secretion in leukocytes, and plays a critical role in blastocyst-endometrium adhesion in the endometrium during the window of implantation (WOI) via interaction with αvβ integrin.
OPN is implicated in cancer progression, neoplastic transformation, and tissue remodeling across several cancers, notably ovarian, colorectal, and hepatocellular carcinoma (HCC). In ovarian cancer, it promotes tumor progression by enhancing cell migration, invasion, and survival, with elevated levels linked to poor prognosis and metastasis. In colorectal cancer, OPN supports tumor growth through angiogenesis and immune evasion, correlating with advanced disease and reduced survival rates. In HCC, it facilitates tumor growth and resistance to apoptosis, where high expression indicates poorer outcomes. OPN also regulates angiogenesis, cell adhesion, apoptosis, and inflammation, which impact cell attachment, migration, invasion, and proliferation across these cancers. Beyond oncology, OPN plays essential roles in immune response regulation, particularly by recruiting immune cells and promoting cytokine secretion in chronic inflammatory conditions. It has emerged as a biomarker for early Alzheimer’s disease due to its influence on neuroinflammation and neuronal survival through microglial activity.
Additionally, OPN is crucial for bone health, facilitating communication between osteoclasts and osteoblasts to regulate bone resorption and formation. Furthermore, OPN contributes to vascular calcification by promoting inflammation and plaque formation in atherosclerosis, advancing heart disease. Expressed by immune cells, it impacts calcification and may serve as a therapeutic target.These diverse functions underscore OPN’s potential as both a therapeutic target and biomarker in a range of conditions, including cancer, chronic inflammation, neurodegeneration, and reproductive diseases. Moreover, OPN’s roles in cell migration, attachment, and invasion suggest a potential link to the pathogenesis of endometriosis through mechanisms similar to those observed in cancer, indicating that OPN could serve as a valuable biomarker for diagnosing or monitoring endometriosis.
Comment 3:
Tables 1 and 2: The accumulation of numbers makes the tables difficult to read (e.g., "Age [yr] (M ± SD; N) 31.90 ± 4.50; 48"). I suggest listing the numbers (n) separately for better clarity.
Response 3:
The table has been changed.
Comment 4:
Additionally, in Table 1, the meaning of the variable "Presence of endometrioma 27 ± 55.10" is unclear without further explanation. Does it refer to the diameter of the endometrioma, the age of the patients, or the percentage of patients with endometrioma?
Response 4:
It refers to the percentage of ednometrioma, this information was add to the table.
Comment 5:
The numbers for controls in the rows "age" (n = 38) and "menstrual cycle" (n = 40) are inconsistent. If data are missing, please indicate this as "N/A."
Response 5:
Thank you for pointing this out. We add N/A, where data was missing.
Comment 6:
Table 3: Abbreviations are used without explanation (e.g., -95% PU, +95% PU). Please provide definitions for all abbreviations.
Response 6:
Definitions for all abbrevaitions were added.
Comment 7:
Was the rAFS staging system the only classification used? The #Enzian classification is currently the most widely used and comprehensive system for endometriosis in Europe. If this information is available, please include the #Enzian scoring for patient characteristics.
Response 7
In our study, we opted not to use the #Enzian classification system primarily due to the limited number of cases involving deep infiltrating endometriosis. The rAFS staging system was more appropriate for our patient cohort, as it provided a clearer framework for the types of endometriosis we encountered. Furthermore, while the #Enzian classification is indeed comprehensive and widely used in Europe, our focus was on the specific characteristics and treatment outcomes of our patient population. The rAFS system allowed us to effectively categorize and analyze the data relevant to our study's objectives. We acknowledge the value of the #Enzian classification and appreciate its utility in broader contexts, but our decision was guided by the specifics of our case load and research goals.